Establishment and evaluation of a circAdpgk-0001 knockdown method using CRISPR–Cas13d RNA-targeting technology

Huang Sijia 1
Qin Hailan 1
Dai Bingxin 1 2
Liu Miao iammiaoliu@126.com 1
Shen Jijia shenjijia@hotmail.com 1
1 Department of Microbiology and Parasitology, Anhui Province key Laboratory of Zoonoses, School of Basic Medical Sciences, Anhui Medical University , Hefei , China
2 Department of Clinical Laboratory, South China Hospital, Medical School, Shenzhen University , Shenzhen , China
Brygadyrenko Viktor
Electronic publication date: 2025 Oct 1
Publication date: 2025
Volume: 13
Electronic Location ID: e20123
Received 2025 Mar 14; Accepted 2025 Sep 1
Copyright: ©2025 Huang et al.
Copyright year: 2025
Copyright holder: Huang et al.
License: This is an open access article distributed under the terms of the Creative Commons Attribution License, which permits unrestricted use, distribution, reproduction and adaptation in any medium and for any purpose provided that it is properly attributed. For attribution, the original author(s), title, publication source (PeerJ) and either DOI or URL of the article must be cited.
License URL: https://creativecommons.org/licenses/by/4.0/

Keywords: CRISPR/Cas13, CircRNAs, RNA targeting, RNA interference

Funding: The National Natural Science Foundation of China No. 82072305 This work was supported by a grant from the National Natural Science Foundation of China (No. 82072305). The funders had no role in study design, data collection and analysis, decision to publish, or preparation of the manuscript.

==============================
Background

The small interfering RNA (siRNA) method has been used to knock down circular RNAs (circRNAs). However, issues such as low efficiency and off-target effects have become increasingly recognized. Recent studies have demonstrated that CRISPR-Cas13 can specifically target and cleave RNA. In this study, we established a CRISPR-Cas13d-based RNA-targeting method to specifically knock down circRNAs, such as circAdpgk-0001, and compared its performance with the siRNA method.

Methods

Four clustered regularly interspaced short palindromic repeats (CRISPR) RNAs (crRNAs) of different nucleotide lengths spanning the back-splicing junction (BSJ) of circAdpgk-0001 were designed. A CRISPR-RfxCas13d plasmid capable of specifically cleaving circAdpgk-0001 was constructed and transfected into the JS-1 cell line. Knockdown efficiency was assessed using quantitative real-time PCR (qRT-PCR) and compared with that of the siRNA method. The expression of activation-related factors alpha-smooth muscle actin (α-SMA) and collagen I in JS-1 cells was further evaluated using qRT-PCR and Western blot.

Results

CRISPR-Cas13d with a 24-nucleotide crRNA showed the highest knockdown efficiency (∼50%). After further optimization, the knockdown efficiency of CRISPR-Cas13d reached 70%, significantly higher than that of the siRNA method (40%). Knockdown of circAdpgk-0001 using Cas13d reduced the expression of collagen I and α-SMA by approximately 40%, which was greater than the reduction achieved by siRNA-mediated knockdown.

Conclusion

CRISPR-Cas13d demonstrated higher efficiency than the siRNA method in knocking down circRNAs, providing a promising tool for investigating circRNA functions.

Introduction

Circular RNAs (circRNAs) are a class of noncoding RNAs, characterized by a single-stranded, closed circular structure, lacking a 5′ cap and a 3′ poly-A tail (Qu et al., 2015). It has been reported that circRNAs exhibit high stability, compared with linear mRNAs, are resistant to degradation, and possess specific backsplice sites. Most circRNAs are localized in the cytoplasm, while a few are found in the nucleus (Huang et al., 2020). An increasing number of studies have shown that circRNAs are involved in the development of various disease (Kristensen et al., 2022; Liu & Chen, 2022). For example, circIRF2 suppresses carbon tetrachloride-induced liver fibrosis by facilitating the nuclear translocation of FOXO3 (Chen et al., 2023b). CircDICAR acts as a novel endogenous regulator in diabetic cardiomyopathy (Yuan et al., 2023). To date, circular RNAs have been demonstrated to possess diverse biological functions, including: (1) miRNA sponge. CircRNAs can sponge miRNAs, thereby relieving miRNA-mediated repression of target genes and indirectly regulating gene expression (Panda, 2018). (2) Modulation of gene transcription. CircRNAs could interact with RNA-binding proteins (RBPs) to form circRNA-RBP complexes that bind to promoter regions, influencing the binding of transcription factors to the promoters and subsequently regulating transcriptional activity (Huang et al., 2020; Zhou et al., 2020). (3) Protein-coding capacity. Recent studies have revealed that circRNAs could recruit ribosomes through specialized mechanisms to undergo translation, producing functional peptides or proteins that might possess unique biological activities (Chen et al., 2023a; Liang et al., 2017).

The CRISPR-Cas systems have grown to be a powerful and widespread biotechnological tool in the fields of molecular biology and genetics, which are divided into two classes and six types. Class 2 systems include types II, V, and VI, in which Cas9, Cas12, and Cas13, respectively, function as single, multidomain effectors. Unlike Cas9 and Cas12, which target DNA, Cas13 targets RNA, including Cas13a, Cas13b, Cas13c, and Cas13d (Wang et al., 2019). Cas13 associates with clustered regularly interspaced short palindromic repeats (CRISPR) RNAs (crRNAs), and cleaves RNA target (Zhang et al., 2019). Compared with the specific small interfering RNA (siRNA) method, Cas13 offers advantages such as convenient design, fewer off-target effects, and highly efficient and specific RNA knockdown in mammalian cells (Chen et al., 2020). Currently, studies have shown that Cas13 has been applied in targeted tumor therapy and viral nucleic acid detection (Liang et al., 2024; Nguyen, Zhang & Pandolfi, 2020). Moreover, Cas13 has been used for gene therapy applications, particularly for the central nervous system (McCallister et al., 2025; Morelli et al., 2023; Powell et al., 2022; Zeballos et al., 2023).

Recently, the widespread application of high-throughput sequencing has led to the discovery of many novel circRNAs in mammalian cells (Salzman et al., 2013). We previously performed high-throughput circRNAs sequencing on primary HSCs isolated from mouse liver fibrosis model induced by Schistosoma japonicum. Then, we were ready to explore the function and mechanism of high-expressed circRNAs. However, in subsequent in vitro cell experiments, the knockdown efficiency of some circRNAs was unsatisfactory using the siRNA technology. According to the literatures, the CRISPR-Cas13 system can be used to screen a group of functional circular RNAs (Li et al., 2021; Li, Wu & Chen, 2022). These reports showed that Cas13 could effectively discriminate circRNAs from mRNAs using crRNAs that target sequences spanning BSJ sites in RNA circles, with high efficiency, specificity, and generality. In this study, we have applied this method to achieve higher knockdown efficiency of circAdpgk-0001 by adjusting the transfection conditions and reagents. Our data have demonstrated that CRISPR-Cas13d could be used to knock down circRNAs with higher efficiency than siRNAs, providing a new tool for the functional study of circRNAs.

Materials and Methods

Cell culture

The JS-1 cell line, a mouse-derived hepatic stellate cell (HSC), was purchased from the Otwo Biotech, China. The cell line was routinely cultured in Dulbecco’s Modified Eagle’s Medium (DMEM) (Gibco, Beijing, China) supplemented with 10% FBS (ExCell, Shanghai, China) at 37 °C with 5% CO2.

CRISPR RNAs (crRNAs) design

The mature sequence was obtained according to circAdpgk-0001 (circAtlas, chr9:59297526—59303551). And the BSJ sites of circAdpgk-0001 were analyzed by Circprimer2.0. Then, 11, 12, 14, and 16 nucleotides were taken respectively on both sides centered on BSJ (BSJ was not included), resulting in crRNAs with lengths of 24, 26, 30, and 34 nucleotides. Finally, based on the principle of base complementary pairing, SnapGene was used to design forward complementary single oligonucleotide chain and reverse one for crRNAs with different lengths for subsequent use.

Plasmid construction and siRNA synthesis

Two µg CRISPR-RfxCas13d plasmid (Addgene, Watertown, MA, USA), two µL BsmBI-v2 enzyme (NEB, USA), five µL 10 × NEBuffer3.1 and water were mixed in a tube to form a 50 µL enzyme digestion system and digested at 55 °C for 1 h. The enzyme digestion products were purified using PCR product purification kit (UEBio, Shanghai, China), and the linearized CRISPR-RfxCas13d vector was obtained. Nine µL of 100 µM forward single oligonucleotide chain, nine µL of 100 µM reverse single oligonucleotide chain and two µL of NEBuffer 2.1 were mixed in a tube. Annealing products were obtained by placing them in a PCR instrument and gradually decreasing from 95 °C to 22 °C at a rate of 1 °C per minute. Three µL annealing product, 50 ng linearized CRISPR-RfxCas13d plasmid, one µL T4 DNA Ligase (Takara, Japan), and one µL 10 × Ligation buffer were mixed in a tube, which was supplemented to 10 µL with water and incubated at 16 °C for overnight. After transformation, individual E. coli DH5α colony in the Petri dishes (Ampicillin) was picked up and moved to a tube containing five ml LB. Place the tube in a shaking incubator at 180 rpm for 6–8 h at 37 °C. The bacterial solution was added to 200 ML LB in a shaking incubator at 180 rpm for 14–16 h at 37 °C for expanded culture. Plasmids were extracted using the E.Z.N.A.® Endo-free Plasmid Maxi Kit (OMEGA Biotek, , Norcross, GA, USA). Then the plasmid was further confirmed through sequencing. siRNAs of circAdpgk-0001 were synthesized to target the junction site of circAdpgk-0001. Three siRNAs were designed and synthesized by Genepharma and another three siRNAs were designed and synthesized by GeneseedBio. In the experiment, NC-siRNA was used as the control group. All sequences of siRNA used in this study were listed in Table S1.

Cell transfection

JS-1 cells were transfected using Lipofectamine®3000 (Thermo Fisher Scientific, Waltham, MA, USA) or jetPRIME Transfection (Polyplus, Illkirch, France) according to the manufacturer’s protocol. Different transfection conditions were designed to improve the knockdown efficiency. Condition 1: after cells were seeded and reached 70–90% confluent, they were stimulated with TGF-β (10 ng/ml) (Novoprotein, Suzhou, China) for 24 h, Lipofectamine®3000 was used for transfection for 24 h. Condition 2: after cells were seeded and reached 70–90% confluent, they were stimulated with TGF-β (10 ng/ml) for 24 h, Lipofectamine®3000 was used for transfection for 48 h. Condition 3: after cells were seeded and reached 70–90% confluent, they were stimulated with TGF-β (10 ng/ml) for 24 h, jetPRIME Transfection was used for transfection for 24 h. Condition 4: after cells were seeded and reached 70–90% confluent, Lipofectamine®3000 was used for transfection and TGF-β was added at the same time for 24 h. Knockdown efficiency of circAdpgk-0001 was confirmed using qRT-PCR.

Quantitative real time PCR (qRT-PCR)

Total RNA extraction from cells was performed with TRIzol reagent (TransGen, Beijing, China). The RNA concentration was measured using a NanoDrop 2000 (Thermo Fisher Scientific, USA). The entire process was carried out under conditions free from RNase contamination. Complementary DNA was synthesized using Evo M-MLV Reverse Transcriptase kit AG11706 (AGbio, Beijing, China). SYBR Green Premix Pro Taq HS qPCR kit AG11701 (AGbio) and CFX96 RT-PCR system (Bio-Rad, USA) were used for qRT-PCR detection. The two-step method was adopted, which included pre-denaturation at 95 °C for 30 s, followed by 40 cycles of denaturation at 95 °C for 5 s and annealing/extension at 60 °C for 60 s. Three replicate wells were set for each sample. GAPDH was used to normalize the expression of mRNAs. The relative abundance of expression was calculated using the 2−△△Ct method. The target primer sequences were designed using Premier 6. The sequences of primers used in qRT-PCR were listed in Table S2.

Western blot

Cells were lysed on ice using RIPA buffer (Beyotime, Haimen, China) supplemented with proteinase inhibitor (Beyotime). Total proteins were extracted and measured using a BCA protein assay kit (Beyotime). Equal amount of protein was subjected to SDS-PAGE electrophoresis separation, and transferred onto PVDF membranes, which were blocked in non-fat milk. Then the membranes were cultured with the primary antibody at 4 °C overnight. Afterwards, the secondary antibody was incubated for 1 h at room temperature. The ECL Substrate Kit (Thermo Fisher Scientific, USA) was applied to visualize the target protein bands. Antibodies dilution was as follows: rabbit monoclonal antibody GAPDH (1:3000; Boster), rabbit monoclonal antibody α-SMA (1:5000, Abcam, USA), rabbit polyclonal antibody Collagen I (1:1000, Boster), and HRP-conjugated goat anti-rabbit antibody (1:10000; ZSGB, Beijing, China). Followed with quantifying by Image J software (NIH, Bethesda, MD, USA) and normalized to GAPDH, the internal control.

Statistical analysis

All data analysis was performed using GraphPad Prism v7.01 software (GraphPad, La Jolla, CA, USA). The results were exhibited as the mean ± standard deviation. Independent sample t-test was used for comparisons between two experimental groups, and one-way analysis of variance (ANOVA) was applied to analyze the difference among multiple groups. Data are mean ± SEM of at least three independent experiments. The p-value < 0.05 was considered significant.

Results

The knockdown efficiency of circAdpgk-0001 by siRNAs under different transfection conditions

To knock down circAdpgk-0001, we designed and synthesized six siRNAs targeting circAdpgk_0001 by two biotechnology companies. After transfecting siRNA-1 through siRNA-6 into the JS-1 cell line, only siRNA-5 showed higher knockdown efficiency, reducing circAdpgk-0001 levels by approximately 30% under transfection condition 1 (Fig. 1A), which was insufficient for subsequent experiments. Therefore, to further improve the knockdown efficiency, we designed different experimental conditions by modifying three factors: prolonging the transfection time, changing the transfection reagent, and co-transfection with TGF-β. The knockdown efficiency under transfection condition 2 was approximately 20% (Fig. 1B). The knockdown efficiency under transfection condition 3 was approximately 40% (Fig. 1C). The knockdown efficiency under transfection condition 4 was approximately 30% (Fig. 1D). Overall, the siRNAs were able to knock down circAdpgk-0001 by approximately 40% under the optimal experimental conditions. circ-Cyp4f18_0001 was knocked down by approximately 30%, and circ-Cdkn3_0002 by approximately 20% using siRNAs under the optimal conditions (Figs. S1A–S1B).

Figure 1 The knockdown efficiency of different siRNAs under various transfection conditions.

(A) The knockdown efficiency of different siRNAs (n = 3), (B–D) The knockdown efficiency under various transfection conditions (n = 3). ns (not significant), * p < 0.05, ** p < 0.01, *** p < 0.001.

Construction of CRISPR-Cas13d plasmid targeting circAdpgk-0001 and its knockdown efficiency

Four crRNAs targeting the BSJ of circAdpgk-0001, with nucleotide lengths of 24, 26, 30, and 34, were designed (Fig. 2A). Four recombinant plasmids targeting circAdpgk-0001 were generated, named Cas13d/24, Cas13d/26, Cas13d/30, and Cas13d/34, with the empty CRISPR-RfxCas13d plasmid serving as a control. All plasmids were sequenced to confirm the accuracy of their nucleotide sequences. Under transfection condition 1, the control plasmid did not affect the expression of circAdpgk-0001, while Cas13d/24 achieved the knockdown efficiency of approximately 50% (Fig. 2B). Prolonged transfection under condition 2 resulted in the knockdown of circAdpgk-0001 by approximately 50% with Cas13d/24 (Fig. 2C). After applying transfection condition 3 and changing the transfection reagent, Cas13d/24 resulted in the knockdown of circAdpgk-0001 by approximately 30% (Fig. 2D). Under transfection condition 4, after cells were simultaneously transfected with TGF-β, Cas13d/24 resulted in the knockdown of circAdpgk-0001 by approximately 70% (Fig. 2E). Furthermore, it was found that the knockdown of circAdpgk-0001 by Cas13d/24 had no effect on the expression of the maternal gene ADPGK (Fig. 2F). These results demonstrated that CRISPR-Cas13d RNA targeting technology could effectively knock down circAdpgk-0001 with high efficiency and specificity.

Figure 2 Construction of CRISPR-Cas13d recombinant plasmids and its knockdown efficiency.

(A) Lengths and sequences of crRNA spacers flanking the BSJ site of circAdpgk-0001; bases flanking the BSJ site are shown in brown. (B) The knockdown efficiencies of each CRISPR-RfxCas13d-crRNA recombinant plasmid (n = 3). (C–E) The knockdown efficiencies of Cas13d/24 under different transfection conditions (n = 3). (F) Expression of the maternal gene ADPGK after the knockdown of circAdpgk-0001 (n = 3). ns (not significant), *p < 0.05, **p < 0.01, ***p < 0.001.

Higher knockdown efficiency of CRISPR-Cas13d, compared with siRNAs

Cell experiments were conducted using the optimal transfection conditions for siRNA-5 and the Cas13d/24 plasmid to compare the knockdown efficiency simultaneously. The knockdown efficiency of the Cas13d/24 recombinant plasmid was significantly higher than that of siRNA-5 (Figs. 3A–3B). The effect of circAdpgk-0001 knockdown on activation-related factors was further investigated in the JS-1 cell line. qRT-PCR results showed that the expression of activation-related factors, collagen I and α-SMA, decreased by approximately 30% after circAdpgk-0001 knockdown with siRNA-5 (Fig. 3C). The expression of collagen I and α-SMA decreased by approximately 40% after circAdpgk-0001 knockdown with Cas13d/24 (Fig. 3D). We compared the effects of the two methods on collagen I and α-SMA and found that the effects were more significant after Cas13d/24 knockdown of circAdpgk-0001 (Fig. 3E). Furthermore, Western blot results revealed that circAdpgk-0001 knockdown also reduced the protein expression of collagen I and α-SMA (Figs. 4A–4B). Moreover, the effect of the Cas13d/24 recombinant plasmid on the protein level of α-SMA was also higher than that of siRNA-5 (Fig. 4C). These results indicated that CRISPR-Cas13 RNA targeting technology could significantly knock down circAdpgk-0001 and reduce the expression of activation-related factors in the JS-1 cell line.

Figure 3 The comparison of knockdown efficiency and effects on mRNA expression levels of collagen I and α-SMA between CRISPR-Cas13d and siRNAs.

(A) The knockdown efficiency of the Cas13d/24 recombinant plasmid and siRNA-5, respectively (n = 3), (B) Comparison of the knockdown efficiency between Cas13d/24 recombinant plasmid and siRNA-5 (n = 3), (C) and (D) Expression of Collagen I and α -SMA mRNA levels after circAdpgk -0 001 knockdown by siRNA-5 and Cas13d/24 (n = 3), (E) Comparison of the effects of two knockdown methods on collagen I and α-SMA (n = 3). ns (not significant), * p < 0.05, ** p < 0.01, *** p < 0.001.

Figure 4 The effect on protein expression levels of collagen I and α-SMA with of CRISPR-Cas13d and siRNAs.

(A and B) Expression of collagen I and α-SMA protein levels after circAdpgk - 0001 knockdown by siRNA-5 and Cas13d/24 (n = 3), (C) Comparison of the effects of two knockdown methods on collagen I and α-SMA at the protein level (n = 3). ns (not significant), * p < 0.05, ** p < 0.01, *** p < 0.001.

Discussion

The small interfering RNA (siRNA) is a process through which target mRNA is degraded, resulting in genetic expression silencing. The transient introduction of exogenous siRNA produces fast and revisable gene silencing as a low-cost, efficient and accurate tool to knock down target RNAs (Alshaer et al., 2021). In contrast to the siRNA method, the CRISPR-Cas13 system is an emerging technology that uses crRNA to guide the Cas13 protein to target single-stranded RNA, including Cas13a, Cas13b, Cas13c, Cas13d, and the newly discovered Cas13X/Y (Cheng et al., 2023). Once activated by the crRNA bearing complementary sequences to the target RNA, Cas13 cleaves the target RNA (Abudayyeh et al., 2017; East-Seletsky et al., 2016).

Among the reported Cas13 family proteins with RNA cleavage activities, Cas13d is the most effective effector, exhibiting the highest knockdown efficiency and no detectable effect on their cognate mRNAs (Konermann et al., 2018; Li, Wu & Chen, 2022). This feature was also confirmed in our study, and the knockdown of circAdpgk-0001 had any no effect on its maternal gene ADPGK. In this study, we also found that the application of CRISPR-RfxCas13d recombinant plasmid to knockdown circAdpgk-0001 resulted in greater stability than that of siRNA. This might be attributed to the fact that the structure of the plasmid is more stable compared to that of siRNA. It has been reported in the literature that the efficiency of back-splicing is much lower, less than 1%, than that of canonical splicing (Liu & Chen, 2022; Zhang et al., 2016). Once circRNA is cleaved by Cas13d, the newly generated circRNA cannot efficiently compensate for the loss, leading to a continuous reduction of circRNA (Li, Wu & Chen, 2022). In addition, the structure of circRNA is more stable and more rigid than linear RNAs (Liu et al., 2019). When circRNA is recognized by crRNA, Cas13d protein can continuously target and cleave the rigid structure in circRNA. However, linear RNAs have a highly dynamic and flexible folding structure, and it is difficult for Cas13d to continuously and effectively target linear RNA (Li, Wu & Chen, 2022).

The key part of CRISPR-Cas13 applications is to design crRNAs with high targeting efficiency and specificity. On the one hand, it maximized the efficiency of the knockdown; on the other hand, it helped to avoid the potential off-target effects, such as incidental cleavage of nearby linear RNA (Li, Wu & Chen, 2022). In addition to requiring perfect base pairing with the target RNA, the length of crRNA is also very important. According to literature reports, the minimum length required for efficient circRNA cleavage induced by Cas13d is 18 nucleotides (nt), and higher circRNA knockdown efficiency is observed using 22 nt or longer lengths, but the knockdown efficiency could not be further improved beyond 30 nt (Li et al., 2021; Zhang et al., 2018). It was also reported that the optimal crRNA length (26 nt) overlapped with the linear cognate mRNA as short as 13 nt, and theoretically such a short gRNA crRNA could not enable to bring Cas13d effector to target linear mRNAs (Konermann et al., 2018; Li et al., 2021). In this study, we designed and synthesized 24 nt, 26 nt, 30 nt and 34 nt crRNAs. The results showed that 24 nt had the highest knockdown efficiency, while 34 nt had almost no effect on circAdpgk-0001 knockdown. It was found that a nuclear localization signal (NLS) could be fused to Cas13d. NLS-fused Cas13d showed a powerful ability to knock down target mRNA with high specificity and high efficiency (∼96%) (Konermann et al., 2013). In order to further improve the knockdown efficiency, we also used the CRISPR-Cas13d of NLS fusion, and constructed the plasmid targeting circAdpgk-0001 with the same method. Unfortunately, the results indicated that the fusion of NLS did not improve the knockdown efficiency of circAdpgk-0001 (data not shown).

CRISPR-Cas13 systems have recently been used for targeted RNA degradation in various organisms. However, collateral degradation of bystander RNAs has limited their applications. To avoid this issue, there have been reports in the literatures, RfxCas13d in vitro and in vivo did not detect collateral RNA degradation (Konermann et al., 2018; Morelli et al., 2023; Wessels et al., 2020), showing higher specificity. More importantly, as reported in the recent article (Tong et al., 2023), several Cas13 variants including Cas13d and Cas13X exhibited efficient on-target activity but markedly reduced collateral activity. Compared with Cas13d and Cas13X, high-fidelity Cas13 variants showed similar RNA knockdown activity to wild-type Cas13 but no detectable collateral damage. Our results indicated that compared with the siRNA method, CRISPR-Cas13d showed higher circRNA knockdown efficiency and provided a new technology for the functional study of circRNAs. Based on these data, we would further identify effects of the high-fidelity Cas13 variants on targeting of the circRNA in the future.

Supplemental Information

Supplemental Information 1 Different sequences of siRNAs

Supplemental Information 2 Primer pairs used in the current study

Supplemental Information 3 The knockdown efficiency of circ-Cyp4f18_0001 and circ-Cdkn3_0002 using the siRNA method

(A) The knockdown efficiency of circ-Cyp4f18_0001 by different siRNAs (n = 3). (B) The knockdown efficiency of circ-Cdkn3_0002 by different siRNAs (n = 3). *p < 0.05, **p < 0.01, ***p < 0.001.

Supplemental Information 4 The raw data for Figure 1

The qRT-PCR raw data shows differential knockdown efficiencies among six siRNAs and across three distinct transfection conditions (n = 3).

Supplemental Information 5 The raw data for Figure 2

The qRT-PCR raw data shows differential knockdown efficiencies among four CRISPR/Cas13d recombinant plasmids and across three distinct transfection conditions.The effect of CRISPR/Cas13d recombinant plasmid on linear maternal gene expression was also examined (n = 3).

Supplemental Information 6 The raw data for Figure 3

The qRT-PCR raw data shows a comparison of the knockdown efficiency of siRNA and Cas13d/24 under optimal transfection conditions.And compare the expression of Collagen I and α-SMA (n = 3).

Supplemental Information 7 The raw data for Figure 4

The expression of Collagen I and α-SMA at the protein level compared by WB, and the gray value was scanned for statistics (n = 3).

Supplemental Information 8 The raw data for Figure S1

The qRT-PCR raw data shows the knockdown efficiency of circ-Cyp4f18_0001 and circ-Cdkn3_0002 with different siRNAs (n = 3).

Supplemental Information 9 MIQE checklist

Additional Information and Declarations

Competing Interests

Author Contributions

Data Availability

The authors declare there are no competing interests.

Sijia Huang performed the experiments, analyzed the data, prepared figures and/or tables, authored or reviewed drafts of the article, and approved the final draft.

Hailan Qin performed the experiments, analyzed the data, prepared figures and/or tables, and approved the final draft.

Bingxin Dai performed the experiments, analyzed the data, authored or reviewed drafts of the article, and approved the final draft.

Miao Liu conceived and designed the experiments, analyzed the data, authored or reviewed drafts of the article, and approved the final draft.

Jijia Shen conceived and designed the experiments, analyzed the data, authored or reviewed drafts of the article, and approved the final draft.

The following information was supplied regarding data availability:

The raw measurements are available in the Supplementary File.

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
