# Peer review of "Establishment and evaluation of a circAdpgk-0001 knockdown method using CRISPR–Cas13d RNA-targeting technology"

_PeerJ, doi:10.7717/peerj.20123_

## Round 0.1 · original submission · Major Revisions

**Language Note:** The review process has identified that the English language must be improved. PeerJ can provide language editing services - please contact us at [email protected] for pricing (be sure to provide your manuscript number and title). Alternatively, you should make your own arrangements to improve the language quality and provide details in your response letter. – PeerJ Staff

Reviewer 1 ·

Basic reporting

In their manuscript, Huang et al. use Cas13d technology to silence the circular RNA circAdpgk-0001. The authors empiracally determine the optimal spacer sequence length and test various conditions before identifying a system that lowered the circRNA by ~70%. They find the linear mRNA is unaffected by targeting of the circRNA and also show that Cas13d appeared to work more efficiently than siRNAs.

Though the manuscript is brief, the study is generally well described and appears well controlled. I have no major questions about the conclusions of the study. I do however have some minor comments to help improve the accuracy of the text.

1. Cas13d is an RNA targeting technology. The authors should not refer to it as “gene editing” like in lines 197, the title or the abstract.

2. Line 173, the authors should explain the rationale for using TGF-beta for increasing knockdown.

3. Line 185, the authors should state which Cas13d variant they used (RfxCas13d, Cas13X, etc.) and the reasoning for the choice.

4. gRNA should be changed to crRNA for accuracy and consistency with the nomenclature in the field.

5. Cas13 has the capacity to trans-cleave bystander RNAs, which can cause off-target effects. The authors should discuss this limitation and whether it may be contributed to their results (unlikely as they saw no change in the linear ADPGK mRNA).

6. Related to OT effects, Cas13 variants with reduced bystander effects have been described. The authors should cite this manuscript (Hong et al. Nat. Biotechnol. 2023) and discuss whether it could be useful for future studies.

7. Line 57, Cas13d has also been used for gene therapy applications, particularly for the CNS (Powell et al. Sci Adv. 2022, Zeballos Nat. Commun. 2023, Morelli et al. Nat. Neurosci. 2023, McCallister Nat. Commun. 2025). These studies should be cited along with the other examples.

8. Line 58, the reference for the tumor therapy example is missing.

9. There are a number of typos throughout the manuscript, particularly in the abstract. The authors should carefully proof read the text.

Experimental design

No comment.

Validity of the findings

No comment.

·

Basic reporting

In this manuscript the authors novel method of circular RNA knock down using CRISPR/Cas13. The data is indeed valuable since the novel method provides significantly higher efficiency (70%) than the siRNA method (40%).

It would be of a great benefit to the paper if you extend the Introduction section with information of particular circRNA functioning, instead of just pointing out the disease-relation of them - it will actualize the research. E.g., use the information from (Zhou, WY., Cai, ZR., Liu, J. et al. Circular RNA: metabolism, functions and interactions with proteins. Mol Cancer 19, 172 (2020). https://doi.org/10.1186/s12943-020-01286-3) and (Chen, R., Wang, S.K., Belk, J.A. et al. Engineering circular RNA for enhanced protein production. Nat Biotechnol 41, 262–272 (2023). https://doi.org/10.1038/s41587-022-01393-0).

Fig 1 and 2: Why the most valuable comparisons of Control vs experiment are so rarely showed?? Fig 1 have not a single of such comparison – just comparisons between some (why only?) experiments, and Fig. 2 – only F variant shows comparisons vs Control! Additionally – you obviously use different grey-scale colors to show different experimental conditions, but with no legend. And different full spectra colors would clearly do the job better.

Experimental design

L206:208 – So, in first experiments you established that siRNA efficiency was ~40% while Cas13 was ~70%. But here you show only a 10% efficiency difference. How could that be?

Validity of the findings

Overall question: Are you sure that the gRNA used for Cas13 can not be used as siRNA by cells RNAi systems? Could it be that the effect of Cas13 is in fact a combination of Cas13 and RNAi effects? This is critical since in that case – the off-target problem remains. Which is even more crucial since such a parameter wasn’t measured.

I haven’t found in text the examples of better repeatability of Cas13 method mentioned in Abstract and in conclusion.

Additional comments

English in text needs minor corrections - frequent misprints.

---

## Round 0.2 · Minor Revisions

**Language Note:** When you prepare your next revision, please either (i) have a colleague who is proficient in English and familiar with the subject matter review your manuscript, or (ii) contact a professional editing service to review your manuscript. PeerJ can provide language editing services - you can contact us at [email protected] for pricing (be sure to provide your manuscript number and title). – PeerJ Staff

Reviewer 1 ·

Basic reporting

The authors have addressed my concerns. I have no additional comments to make. The conclusions of the manuscript are supported by the results, in my view.

Experimental design

-

Validity of the findings

-

·

Basic reporting

-

Experimental design

With all the respect to the authors, you’ve added only comparisons of Control with the first experimental condition. Each comparison needs to be clearly stated - which variants are compared? In other words, you need to add lines connecting each of the experimental variants with the Control value with captions indicating the significance level (ns,*,**, etc), or only the statistically significant ones, with a statement, e.g., “only the statistically significant comparisons are shown”.

Validity of the findings

-

---

## Round 0.3 · accepted · Accept

Dear Dr. Shen, I congratulate you on the acceptance of this article for publication.

·

Basic reporting

pass

Experimental design

-

Validity of the findings

-